# A Large Sample Retrospective Study on the Distinction of Voriconazole Concentration in Asian Patients from Different Clinical Departments

**DOI:** 10.3390/ph14121239

**Published:** 2021-11-29

**Authors:** Yichang Zhao, Chenlin Xiao, Jingjing Hou, Jiamin Wu, Yiwen Xiao, Bikui Zhang, Indy Sandaradura, Miao Yan

**Affiliations:** 1Department of Pharmacy, The Second Xiangya Hospital, Central South University, Changsha 410011, China; zhaoyichang@csu.edu.cn (Y.Z.); xiaochenlin@csu.edu.cn (C.X.); houjingjing2020@csu.edu.cn (J.H.); wujiamin8@foxmail.com (J.W.); xiaoyw2005@163.com (Y.X.); bikui_zh@126.com (B.Z.); 2School of Medicine, University of New South Wales, Sydney, NSW 2052, Australia; Indy.Sandaradura@health.nsw.gov.au; 3Centre for Infectious Diseases and Microbiology, Westmead Hospital, Sydney, NSW 2145, Australia

**Keywords:** voriconazole, different departments, different population, trough concentration, therapeutic drug monitoring

## Abstract

Voriconazole (VRZ) is widely used to prevent and treat invasive fungal infections; however, there are a few studies examining the variability and influencing the factors of VRZ plasma concentrations across different clinical departments. This study aimed to evaluate distinction of VRZ concentrations in different clinical departments and provide a reference for its reasonable use. From 1 May 2014 to 31 December 2020, VRZ standard rates and factors affecting the VRZ trough concentration were analyzed, and a multiple linear regression model was constructed. The standard rates of VRZ in most departments were above 60%. A total of 676 patients with 1212 VRZ trough concentrations using a dosing regimen of 200 mg q12h from seven departments were enrolled in the correlation analysis. The concentration distribution varied significantly among different departments (*p* < 0.001). Fifteen factors, including department, CYP2C19 phenotype, and gender, correlated with VRZ concentration. A multiple linear regression model was established as follows: VRZ trough concentration = 5.195 + 0.049 × age + 0.007 × alanine aminotransferase + 0.010 × total bilirubin − 0.100 × albumin − 0.004 × gamma-glutamyl transferase. According to these indexes, we can predict possible changes in VRZ trough concentration and adjust its dosage precisely and individually.

## 1. Introduction

The use of broad-spectrum antibiotics, glucocorticoids, immunosuppressants, and invasive diagnostic and therapeutic procedures increase the risk of invasive fungal infections (IFIs). Organ transplant recipients [1], cancer patients [2], and critically ill patients [3] are more likely to suffer from IFIs. Rates of antibiotic resistance are also increasing; therefore, prevention and treatment are becoming increasingly important. Voriconazole (VRZ) is a second-generation triazole antifungal that is the first-line treatment for invasive aspergillosis [4]. Its broad coverage [5,6] includes *Candida*, *Aspergillus*, *Fusarium*, and *Cryptococcus neoformans*.

Several guidelines regarding VRZ dosing and individual departments report unique experiences with the drug [7]. CYP2C19 metabolizes VRZ, and its pharmacokinetics are affected by the CYP2C19 genotype [8,9]. The CYP2C19 gene polymorphism and its mediated metabolism result in significant large individual variations in the pharmacokinetic parameters of voriconazole, such as apparent volume of distribution in patients. Using the standard dosing regimen, the trough concentration of VRZ ranges from 0.2 μg/mL to 12 μg/mL [10,11]. Studies showed a significant exposure–response relationship between the trough concentrations of VRZ, clinical efficacy, and adverse events [12,13]. VRZ has a narrow therapeutic window and is associated with adverse reactions [14,15,16,17] such as neurotoxicity, hepatotoxicity, visual impairment, and skin cancer. For these reasons, therapeutic drug monitoring (TDM) of VRZ is of the utmost importance for its safe and effective use.

There have been studies of the pharmacokinetics of VRZ in kidney transplant recipients [18], cancer patients [19], and children [20]. Although VRZ is used in many departments, there are no studies on the use of VRZ among departments, specifically concerning differences in VRZ concentration. To address these issues, this study aimed to record VRZ concentrations in several clinical departments to provide a reference regarding its rational use.

## 2. Results

### 2.1. The Overall Standard Rate of VRZ Trough Concentration

During 1 May 2014, and 31 December 2020, we collected 5904 VRZ trough concentrations from 2484 patients from 28 departments in our hospital. All patients were Asians. In order to confirm that the results reflect the facts, we excluded referral bias. The patients treated with VRZ in this hospital came from seven departments, including the urological organ transplantation and infectious diseases departments. To analyze the overall standard rate which means percentage of samples achieving the target range (1.0 to 5.5 μg/mL), we included 5388 blood samples from these seven departments. The distribution of overall VRZ trough concentrations is shown in Figure 1. The standard rate of VRZ concentration was highest in the urological organ transplantation department, 73.9% (1923/2601). The rates in the hematology, infectious diseases, and pneumology departments were all above 60% [68.9% (597/867), 65% (548/843), and 63.3% (314/496)], respectively. In the pediatric department, the rate was only 41.6% (57/137). By contrast, in 48.9% of the pediatric samples (67/137), the concentration was less than 1 μg/mL. This proportion of low concentration was significantly higher than that of the other six departments. Although the standard rate in the emergency department was more than 50%, the proportion of high concentration samples was 33.9% (105/310), the highest among the seven departments. This finding may be related to the fact that the clinicians working in the emergency department adopt the standard VRZ dosing regimen and rarely carry out TDM to adjust the dose accordingly.

### 2.2. The Standard Rate of VRZ Trough Concentration Using 200 mg q12h

According to the recommendations of the drug instructions, the common medication regimen of voriconazole is 200 mg q12h, which was used by most patients in our hospital. Therefore, we conducted an analysis limiting the VRZ administration regimen up to a maintenance dose of 200 mg q12h. The VRZ concentration distribution is shown in Figure 2 (2183 VRZ trough concentrations). The standard rates of VRZ concentration in the urological organ transplantation and hematology departments were the highest, 76.6% and 70.3%, respectively. This may be related to the cooperation between the pharmacy department of our hospital and these departments regarding TDM and dose adjustment of VRZ. This finding suggests that the use of TDM to adjust the dosage of VRZ improves the standard rate of VRZ. The standard rates in the pneumology and general surgery departments were above 60%; those of the pediatric and general surgery departments exceeded 50% and 60%, respectively, and the proportion of low concentration decreased from 48.9% to 37.1%, from 23.9% to 16.1%, respectively. It is worth noting that the standard rate of the infectious diseases department was as high as 65% before the dosing regimen was limited, but it dropped to 42.4% after limiting. The proportion of high VRZ concentration in this department increased from 28.7% before the limit to 50.0% after the limit. The change range and the high concentration ratio were significantly higher than those of other departments, suggesting that patients in the infectious diseases department may be more likely to accumulate VRZ.

### 2.3. Patient Characteristics

A total of 676 patients with 1212 VRZ trough concentrations using a dosing regimen of 200 mg q12h from seven departments were enrolled in the final analysis. The demographic parameters and primary physiological indicators are listed in Table 1. The median age was 52 years (range, 4–91 years); 70.7% were male; 58 patients received VRZ in the urological organ transplantation department, 180 in the hematology department, 78 in the infectious diseases department, 157 in the department of pneumology, 143 in the emergency department, 21 in the pediatric department, and 39 in the general surgery department.

Total of 454 patients underwent CYP2C19 phenotype detection; 10.6% (*n* = 48) were PM, 47.1% (*n* = 214) were IM, 41.4% (*n* = 188) were EM, and 0.9% (*n* = 4) were UM. The median TDM result of 1212 patients was 3.54 μg/mL (in the target range) and the intraquartile range was 1.76–5.63 μg/mL.

### 2.4. Difference of VRZ Concentration among Seven Departments

VRZ concentrations among seven departments are displayed in Table 2 and Figure 3. The VRZ concentration in the infectious diseases department was 6.29 ± 4.92 μg/mL, significantly higher than that of other departments, and it exceeded the target range. The VRZ concentrations of the other six departments were within the target range. Kruskal-Wallis test revealed significant differences in VRZ concentrations among groups concerning departments (*p* < 0.001). Compared with the urological organ transplantation department, the concentration distributions in the hematology (*p* < 0.001), infectious diseases (*p* < 0.001), pneumology (*p* < 0.001), emergency (*p* < 0.001), and general surgery (*p* = 0.035) departments were significantly different, only the pediatric department (*p* = 1.000) was not.

### 2.5. Determinants of VRZ Trough Concentration

In the univariate analysis, effects gender, age, CYP2C19 phenotypes, departments, and 12 biochemical indexes were studied in enrolled samples. The results are displayed in Table 3. Gender, age, alanine aminotransferase (ALT), aspartate aminotransferase (AST), total bilirubin (TBIL), international normalized ratio (INR), and serum creatinine (CREA) showed significant positive linear trends with VRZ concentration, and the Spearman correlation coefficients (rs) were 0.067 (*p* =0.019), 0.323 (*p* < 0.001), 0.081 (*p* = 0.005), 0.255 (*p* < 0.001), 0.208 (*p* < 0.001), 0.395 (*p* < 0.001), and 0.071 (*p* = 0.014), respectively. Significant negative correlations were observed between CYP2C19 phenotypes (rs = −0.114, *p* = 0.001), red blood cell specific volume (HCT, rs = −0.150, *p* < 0.001), hemoglobin (HGB, rs = −0.154, *p* <0.001), platelets (PLT, rs = −0.165, *p* < 0.001), gamma-glutamyl transferase (GGT, rs = −0.300, *p* < 0.001), alkaline phosphatase (ALP, rs = −0.282, *p* = 0.001), albumin (ALB, rs = −0.254, *p* < 0.001), and VRZ concentration. No correlation was observed between white blood cell (WBC) and concentration of VRZ (*p* = 0.052).

Except for general surgery, the other five departments showed significant differences in VRZ concentration from the urological organ transplantation department for the department dummy variable. The concentration of VRZ in departments of infectious diseases (*p* < 0.001), pneumology (*p* = 0.007), and emergency (*p* < 0.001) was significantly higher than that of urological organ transplantation department. By contrast, there were significant negative correlations between the hematology department and VRZ concentration (*p* = 0.002) and between the pediatric department and VRZ trough concentration (*p* < 0.001).

On the basis of the results of univariate analysis, we constructed a multiple linear regression model using the backward method. The final model revealed that patient age (coefficient [β] = 0.270; *p* = 0.007), ALT (β = 0.179; *p* = 0.080), TBIL (β = 0.290; *p* = 0.004), ALB (β = −0.208; *p* = 0.034), and GGT (β = −0.287; *p* = 0.006) were determinants of VCZ concentration (Figure 4). Details of the optimal multiple linear regression model are presented in Table 4. 

If patient age increased by 1 year, the concentration of VRZ increased by 0.049 μg/mL. VRZ concentration increased by 0.010 μg/mL with one unit increase in TBIL, the concentration tended to be 0.100 μg/mL lower if ALB increased by one unit. With 1 U/L ALT and GGT increases, the concentration increased by 0.007 μg/mL and decreased by 0.004 μg/mL respectively. The linear regression equation was as follows: VRZ trough concentration = 5.195 + 0.049 × age + 0.007 × alanine aminotransferase  + 0.010 × total bilirubin − 0.100 × albumin − 0.004 × gamma-glutamyl transferase(1)

### 2.6. Diagnosis of the Multiple Linear Model

The fitness coefficient of the final regression equation was R^2^ = 0.270, indicating that these factors explained 27.0% of the variability in the disposition of VRZ. The F-value of final regression was 6.982 with a *p*-value of <0.001, suggesting a linear regression relationship among these factors. In addition, the collinearity of age, ALT, GGT, TBIL, and ALB was diagnosed using the variance inflation factor (VIF), and final factors were not collinear with one another (VIF < 2). We finally evaluated the residuals. The residual of the final model established obeyed normal distribution and conformed to the precondition of the regression equation, suggesting that the final model was stable and reliable (Appendix A). Result of bootstrap showed that all coefficients in our final model were within the 95% confidence interval of the corresponding predicted value (Appendix A).

## 3. Discussion

To the best of our knowledge, this study is the first to show that VRZ trough concentrations vary significantly by the department and patient characteristics and provides a reference for VRZ dose adjustment in Asia according to TDM.

The multiple linear regression of 1212 VRZ trough concentrations is the first systematic assessment of factors governing the magnitude of VRZ concentration among different populations. This approach is more reliable than the classic population pharmacokinetics analysis because it is not limited to a specific population. Departments are related to basic diseases and are reflected in different aspects of various indicators which has already been considered in the analysis. Thus, the patient population is not a factor in the final model.

Similar to the findings of many previous studies [8,19,21,22,23,24,25,26,27,28,29,30,31], the model found that VRZ concentration is significantly affected by patient age. Kang et al. found that ALT is a significant factor affecting VRZ concentration, in agreement with our results [32]. Although there is substantial evidence linking AST and VRZ concentration, we found no significant effect of AST in our model [33,34,35]. There are possible explanations for this. Most of the previous studies were limited to specific populations, such as patients with low serum albumin [34] or hematological malignancies [35]. The other reason may be differing pharmacokinetic characteristics such as the clearance and the volume of distribution among populations [36,37], which were reported to have different influence factors such as platelet count, age, and CYP2C19 phenotype.

In addition to ALT and AST, levels of ALP and GGT are reliable parameters for liver function. These also showed an impact on VRZ concentration because VRZ is primarily mainly in the liver. Multiple linear regression analysis showed that the concentration would decrease significantly if GGT increased, suggesting a significant negative correlation between GGT and VRZ trough concentrations. The result is similar to previous findings [19,38] and conforms to the metabolic characteristics of VRZ. It has been reported that ALP had a significant effect on VRZ concentration in kidney transplantation recipients [39] and oncology patients [30]. However, there was no similar result regarding ALP in the present study. This finding might have been due to significant differences in the population, study design differences, or the different methods for constructing the model.

We found that TBIL was a determinant of VCZ concentration, similar to a study of patients undergoing allogeneic hematopoietic stem cell transplantation [40] and critically ill patients [41]. Many researchers [22,23,42] aimed to investigate the relationship between ALB and VRZ trough concentrations, and lower ALB level was found to predict VRZ concentration significantly. Li et al. [22] demonstrated that VRZ concentration in critically ill patients tended to be lower if ALB increased, and our result agrees. The details of other factors affecting VRZ concentration are displayed in Table 5. 

A total of 58.4% of pediatric samples (80/137) did not reach the target range, similar to findings of other researchers [49,50]. Due to minimal studies, TDM of VRZ in Asian children is not available for clinicians, and further research is necessary. Because of age, weight, and other reasons, the commonly used dosage regimen of VRZ in pediatric patients in our hospital included maintenance doses of 80 mg q12h and 100 mg q12h. Of 137 pediatric samples, only 35 (25.5%) used the VRZ regimen of 200 mg q12h, which was the lowest proportion among the seven departments. It is worth noting that the supratherapeutic rate was still as high as 37.1% after the dosage limitation of 200 mg q12h, indicating that VRZ was insufficient in pediatric patients, possibly related to the unique metabolic characteristics of VRZ in children.

Among the 310 samples from the emergency department, 253 (81.6%) were collected with a maintenance dose of 200 mg q12h, indicating that most emergency department patients received VRZ standard administration scheme. The relatively high overall supratherapeutic rate suggests that the emergency department should perform VRZ TDM more often and adjust the maintenance dose according to the monitoring condition changes and clinical reactions. To reduce the incidence of adverse reactions and improve the standard rate of VRZ, individualized treatment needs to be tailored based on liver and kidney function and TDM results.

Only 14.0% (118/843) infectious diseases department samples were collected with a maintenance dose of 200 mg q12h, since VRZ is generally given at 100 or 200 mg qd. Our study showed that the total bilirubin in the infectious diseases department was significantly higher than that of other departments, suggesting that their liver function was impaired. VRZ was more likely to accumulate, possibly explaining the relatively high serum concentration of VRZ (median 5.49 μg/mL). Meanwhile, the reason for high SD in infectious disease patients may be that these patients were mostly accompanied by chronic hepatitis B, hepatitis A, alcoholic hepatitis, etc. There was a great interindividual variability in their liver function. Voriconazole is metabolized by drug metabolizing enzyme CYP2C19 [8,9], and its concentration has a great correlation with liver function [10,11]. In general, we believe that hospital pharmacists should perform TDM to adjust VRZ doses.

Our study should be advanced in the future. In the current study, the numbers of patients in each department and their VRZ sample size distribution were uneven. For example, the hematology department included 300 VRZ concentration points from 180 patients, while the pediatrics only included 35 VRZ samples from 21 children. This is primarily due to the different disease characteristics among departments. Furthermore, many patients did not undergo CYP2C19 genotyping, and the current analysis results may not be sufficient to reflect the overall clinical characteristics. Among the 454 patients who underwent genotyping, only four (less than 1%) had the RM genotype, and no UM patients were found. This finding may be related to the low levels of UM genotype in China. Finally, a retrospective study carries inherent limitations, including the inability to guarantee the integrity of the data.

## 4. Materials and Methods

### 4.1. The Patients and Inclusion Criteria

This retrospective study was performed at the Second Xiangya Hospital of Central South University after approval by the Ethics Committee of the Second Xiangya Hospital of Central South University (ChiCTR.org Registration number: ChiCTR2100048728). The data were collected from May 2014 to December 2020. The ethics committee approved the application for exemption of informed consent because this was a retrospective study without any intervention. Inclusion criteria were as follows: (i) Patients who used VRZ for antifungal treatment in our hospital; (ii) patients with IFI and risk factors for fungal infection were diagnosed, clinically diagnosed, or suspected; (iii) the steady-state blood concentration of VRZ was monitored at least once. We excluded patients from whom we could not obtain accurate information of VRZ dosage or critical clinical data, those who underwent plasma exchange therapy when the blood sample was collected, those known to be allergic to VRZ or any of its components, and those judged by the researcher to be unsuitable for the study. Gender, age, and department are not limited.

### 4.2. VRZ Administration and Data Collection

The dose of VRZ was recorded without intervention. Further analysis of the influencing factors was conducted in patients with a maintenance dose of 200 mg q12h intravenously or orally, according to the recommendations of the drug instructions. Clinicians conducted repeated TDM and adjusted dosage according to the clinical response and TDM results.

Using a standardized data collection form, we extracted the following information from the electronic medical record information system: demographic information (age, gender, weight, and ethnicity); clinical data (primary disease type, infection diagnosis, body temperature, bacterial culture, and drug sensitivity results); laboratory test results (complete blood count, inflammatory index, liver function, and kidney function); CYP2C19 phenotype; and treatment details (administration route, dose, frequency and duration of use, sampling time, duration of infusion, drug concentrations, and concomitant medications).

### 4.3. Blood Sampling and Analytical Assays

Blood was collected 3 days after administering the loading dose of VRZ or 5 days if a loading dose was not administered. We used a fully validated automatic two-dimensional high-performance liquid chromatography technique (Demeter Instrument Co., Ltd., Changsha, China) developed and validated by ourselves before to measure VRZ concentration. An ASTON FRO C18 (100 mm × 3.0 mm, 5 μm, ANAX) column was used for the first-dimensional chromatographic column, and the second was an ASTON HD C18 (150 mm × 4.6 mm, 5 μm, ANAX) column. The intra-day and inter-day precisions were 1.94–2.22% and 2.15–6.78%. The absolute and relative recovery ranged from 88.2% to 93.6% and 94.2% to 105.3%, respectively. The stability of blood sample at room temperature for 8 h and at −20 °C of three repeated freeze-thaw cycles was within ±8% and ±10%, respectively. In addition, our laboratory has passed the external quality assessment, which can ensure the performance of measurement.

### 4.4. CPY2C19 Genotyping and Phenotype Assignment

Blood samples (1–3 mL) from some patients were obtained for CPY2C19 genotype detection. DNA was purified using the E.Z.N.A^®^ SQ Blood DNA Kit II (Omega Bio-Tek, Inc., Norcross, GA, USA) method. We use the Sanger dideoxy DNA sequencing method with the ABI3730XL DNA Analyzer (ABICo.; BioSune Biotechnology Co., Ltd., Shanghai, China) to carry out CYP2C19 genotyping. Based on genotyping results, CYP2C19 phenotypes were classified into five categories according to the definition of the Clinical Pharmacogenetics Implementation Consortium [7]: ultrarapid metabolizer (UM), rapid metabolizer (RM), extensive metabolizer (EM), intermediate metabolizer (IM), and poor metabolizer (PM).

### 4.5. The Standard Rate of VRZ Trough Concentration

Departments with more than 100 VRZ blood samples were included. A trough concentration between 1.0 and 5.5 μg/mL was considered the ideal target range [51,52]. To analyze the distribution of VRZ concentration and its standard rate, the VRZ concentration was divided into three groups: less than 1 μg/mL, 1–5.5 μg/mL, and >5.5 μg/mL. The proportion of group 1–5.5 μg/mL was considered as the standard rate. Based on the overall standard rate analysis, we limited the VRZ dosing regimen to evaluate the difference of VRZ concentration among different departments and the influencing factors in those departments under the same VRZ dosage. TDM results of samples with a maintenance dose of 200 mg q12h were selected, and the VRZ concentration distribution and standard rates were compared with the results before dose limiting.

### 4.6. Differences of VRZ Concentration among Different Departments and the Influencing Factors

We analyzed the serum concentrations in patients whose VRZ dosing regimen was 200 mg q12h using Kruskal–Wallis H test. We used the Bonferroni method to compare distributions across each pair of the department, and the urological organ transplantation department was used as the reference. We searched the recent literature concerning the influencing factors of VRZ plasma concentration and identified 16 factors for correlation analysis. We performed univariate analysis, multiple regression analysis, and the diagnosis of the final model. The department was set as a dummy variable, and also the urological organ transplantation department as the reference.

In the correlation analysis, the missing values were based on the paired exclusion cases. The continuous variables that conform to normal distribution were expressed as mean ± SD, and those that did not were expressed as median (interquartile range). The normality of quantitative data was tested using the Shapiro–Wilk test. Categorical data were expressed as frequency and rate. A two-sided *p*-value < 0.05 was considered statistically significant. To perform the model diagnosis, we performed the goodness of fit test, the test of linearity, and the evaluation of the residual. We used the bootstrap method to construct a validation cohort and verify the multiple linear regression model. We used SPSS 25.0 (IBM Corp, Armonk, NY, USA) software.

## 5. Conclusions

Age, alanine aminotransferase, total bilirubin, albumin, and gamma-glutamyl transferase significantly affect VRZ trough concentration. Concentration may increase with age, alanine aminotransferase, and total bilirubin. By contrast, concentration may decrease with increased albumin and gamma-glutamyl transferase. According to physiological indexes, we can predict possible changes of VRZ trough concentration and adjust the dosage precisely and individually. Carrying out dose optimization to achieve recommended doses for different populations is an ongoing challenge.

## Figures and Tables

**Figure 1 pharmaceuticals-14-01239-f001:**
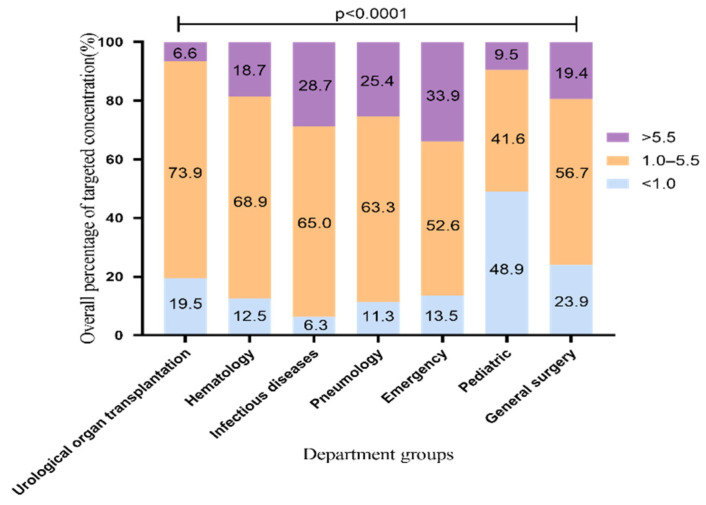
Overall distribution of VRZ concentration (5388 blood samples). The data represent the percentage.

**Figure 2 pharmaceuticals-14-01239-f002:**
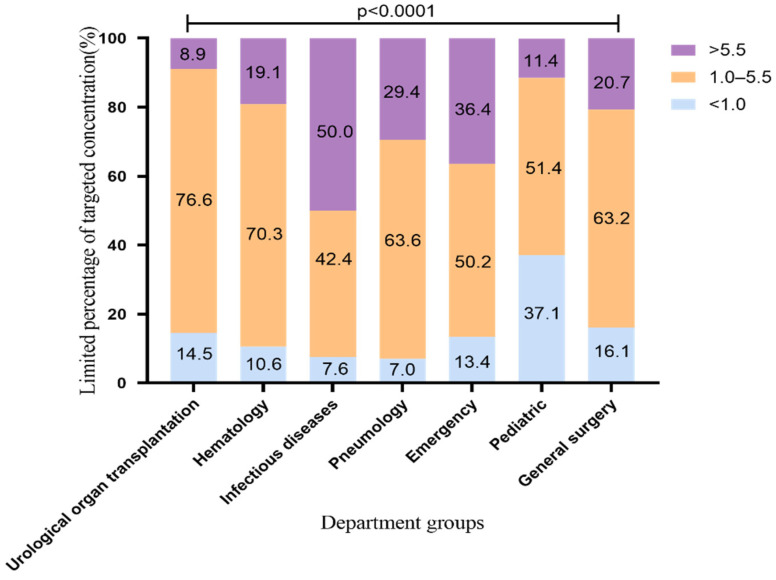
Distribution of VRZ concentration using VRZ 200 mg q12h (2183 blood samples). The data represent the percentage.

**Figure 3 pharmaceuticals-14-01239-f003:**
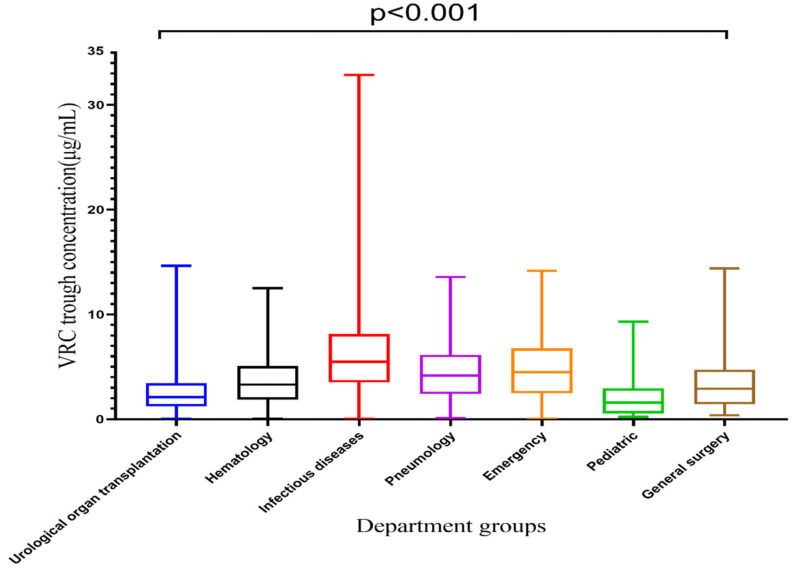
Box plot of VRC trough concentration categorized by department, which is represented by median, minimum, maximum, and interquartile range.

**Figure 4 pharmaceuticals-14-01239-f004:**
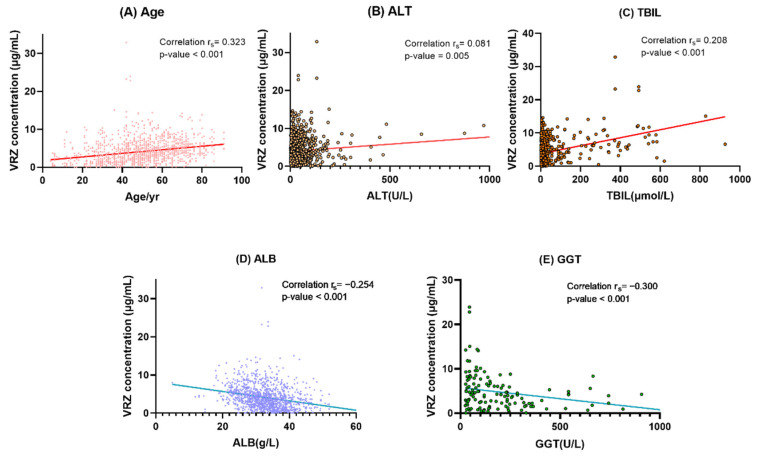
Correlation analysis between VRZ concentration and its determinants. (**A**) Age. (**B**) ALT alanine aminotransferase. (**C**) TBIL total bilirubin. (**D**) ALB albumin. (**E**) GGT gamma-glutamyl transferase. In (**B**,**D**,**E**), a point is omitted outside the coordinate axis.

**Table 1 pharmaceuticals-14-01239-t001:** Clinical data of patients whose administration regimen of voriconazole is 200 mg q12h.

Characteristic	Value	Range
Male, N (%)	478 (70.7%)	
Age (years)	52.0 [40.0–64.0]	4–91
VRZ concentration (μg/mL)	3.54 [1.76–5.63]	0.05–32.86
**CYP2C19 phenotypes**		
Poor metabolizers, N (%)	48 (10.6%)	
Immediate metabolizers, N (%)	214 (47.1%)	
Extensive metabolizers, N (%)	188 (41.4%)	
Rapid metabolizers, N (%)	4 (0.9%)	
**Physiological and biochemical indexes**		
WBC (10^9^/L)	6.05 [3.15–9.47]	0.01–38.58
HCT (%)	27.30 [22.10–33.20]	13.00–60.70
HGB (g/L)	87.0 [71.0–108.0]	39.0–199.0
PLT (10^9^/L)	147.0 [54.0–242.0]	1.0–692.0
ALT (U/L)	20.8 [11.6–42.1]	0.2–1948.2
AST (U/L)	27.2 [16.4–47.6]	1.2–5596.3
GGT (U/L)	123.4 [53.1–246.6]	24.0–1365.1
ALP (U/L)	154.7 [106.4–233.7]	36.1–1231.5
TBIL (μmol/L)	8.4 [5.8–14.7]	1.6–926.6
ALB (g/L)	32.6 ± 6.4	4.9–76.0
INR	1.19 [1.06–1.41]	0.72–9.12
CREA (μmol/L)	68.4 [49.5–105.7]	13.4–1062.4

The normality of quantitative data was analyzed by Shapiro–Wilk normal test, the non-normal distribution was presented by median(IQR), the normal distribution was presented by mean ± SD. ALB, albumin; ALP, alkaline phosphatase; ALT, alanine aminotransferase; AST, aspartate aminotransferase; CREA, serum creatinine; GGT, gamma-glutamyl transferase; HCT, red blood cell specific volume; HGB, hemoglobin; INR, international normalized ratio; IQR, interquartile range; PLT, platelets; SD, standard deviation; TBIL, total bilirubin; WBC, white blood cell.

**Table 2 pharmaceuticals-14-01239-t002:** Voriconazole trough concentration among different hospital departments.

Hospital Departments	Urological Organ Transplantation	Hematology	Infectious Diseases	Pneumology	Emergency	Pediatric	General Surgery	*p*-Value
VRZ concentration (μg/mL, mean ± SD)	2.57 ± 2.07	3.60 ± 2.33	6.29 ± 4.92	4.58 ± 2.77	4.71 ± 2.96	2.24 ± 2.25	3.78 ± 3.11	<0.001

**Table 3 pharmaceuticals-14-01239-t003:** Correlation analysis of voriconazole trough concentration.

Variable	Coefficient Index	*p*-Value
Gender	0.067 *	0.019
Age	0.323 **	<0.001
CYP2C19 phenotypes	−0.114 **	0.001
**Departments**		
Hematology	−0.087 **	0.002
Infectious diseases	0.235 **	<0.001
Pneumology	0.078 **	0.007
Emergency	0.105 **	<0.001
Pediatric	−0.103 **	<0.001
General surgery	−0.026	0.357
**Physiological and biochemical indexes**	27.30 [22.10–33.20]	13.00–60.70
WBC	0.052	0.074
HCT	−0.150 **	<0.001
HGB	−0.154 **	<0.001
PLT	−0.165 **	<0.001
ALT	0.081 **	0.005
AST	0.255 **	<0.001
GGT	−0.300 **	<0.001
ALP	−0.282 **	0.001
TBIL	0.208 **	<0.001
ALB	−0.254 **	<0.001
INR	0.395 **	<0.001
CREA	0.071 *	0.014

* The variables are significant, at the level of 0.05 (double tail); ** the distinction was statistically significant, at the level of 0.01 (double tail).

**Table 4 pharmaceuticals-14-01239-t004:** Multiple linear regression analysis of voriconazole trough concentration determinants.

Variable	Coefficient	T	*p*-Value	VIF
Age	0.049	2.784	0.007	1.043
ALT	0.007	1.772	0.080	1.128
TBIL	0.010	2.990	0.004	1.045
ALB	−0.100	−2.155	0.034	1.032
GGT	−0.004	−2.821	0.006	1.150
Constant value	5.195	2.768	0.007	
F		6.982		
P		<0.001		
R2		0.270		

Dependent variable: voriconazole trough concentration.

**Table 5 pharmaceuticals-14-01239-t005:** Factors affecting voriconazole concentration.

Factors	References	Number of Patients
Age	Tian et al. 2021 [21]	108
	Li et al. 2020 [22]	216
	Mafuru et al. 2019 [19]	113
	Wei et al. 2019 [23]	67
	You et al. 2018 [24]	64
	Allegra et al. 2018 [25]	237
	Shao et al. 2017 [26]	86
	Niioka et al. 2017 [27]	65
	Wang et al. 2014 [8]	151
	Hoenigl et al. 2013 [28]	61
	Choi et al. 2013 [29]	27
	Lombardi et al. 2012 [30]	32
	Dolton et al. 2012 [31]	201
ALT	Kang et al. 2020 [32]	114
AST	Yuan et al. 2020 [33]	193
	Hirata et al. 2019 [34]	42
	Saini et al. 2014 [35]	69
GGT	Cheng et al. 2019 [38]	166
	Mafuru et al. 2019 [19]	113
ALP	Zhao et al. 2021 [39]	93
	Wang et al. 2014 [8]	151
	Saini et al. 2014 [35]	69
	Lombardi et al. 2012 [30]	32
TBIL	Zeng et al. 2020 [40]	244
	Ruiz et al. 2019 [41]	33
	Saini et al. 2014 [35]	69
ALB	Li et al. 2020 [22]	216
	Wei et al. 2019 [23]	67
	Dote et al. 2016 [42]	63
CYP2C19 genotype	Blanco-Dorado et al. 2020 [43]	78
	Yuan et al. 2020 [33]	193
	Mafuru et al. 2019 [19]	113
	You et al. 2018 [24]	64
Concomitant use	Mafuru et al. 2019 [19]	113
	Hu et al. 2018 [20]	42
	Chayakulkeeree et al. 2015 [44]	54
	Kim et al. 2014 [45]	64
INR	Wang et al. 2018 [46]	78
	Lombardi et al. 2012 [30]	32
PLT	Zhao et al. 2021 [39]	93
	Tang et al. 2019 [36]	57
HGB	Zhao et al. 2021 [39]	93
CREA	Allegra et al. 2018 [25]	237
Proinflammatory Cytokines	Mafuru et al. 2019 [19]	113
Obesity	Takahashi et al. 2020 [47]	44
Diarrhea	Nakayama et al. 2020 [48]	44

## Data Availability

Data is contained within the article or Appendix A.

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
