# Peer review of "A Large Sample Retrospective Study on the Distinction of Voriconazole Concentration in Asian Patients from Different Clinical Departments"

_pharmaceuticals, 2021, doi:10.3390/ph14121239_

Round 1

Reviewer 1 Report

The paper is well written and it might be of interest to the readers. Nevertheless there are some issues that need to be clarified. It is not clear to the reviewer if the multiple linear regression model has been tested on a validation cohort to verify the correspondence between the predicted and the measured VRZ values. The analytical metod reported for TDM based on a two-dimensional chromatography is quiite uncommon and might be a source of variability of results. Do the laboratory performs proficiency testing to test their performance?

Author Response

Point 1: The paper is well written and it might be of interest to the readers. Nevertheless there are some issues that need to be clarified.

Response 1: Thank you very much for the recognition of our work and these insightful comments. We will appreciate this opportunity to share our work and have revised the manuscript carefully. According to your suggestion, we have substantially revised the manuscript as follows.

Point 2: It is not clear to the reviewer if the multiple linear regression model has been tested on a validation cohort to verify the correspondence between the predicted and the measured VRZ values.

Response 2:

Thank you very much for your careful review and valuable advice. We have used the bootstrap method to construct a validation cohort and verify the multiple linear regression model. The result showed that all coefficients in our final model were within the 95% confidence interval of the corresponding predicted value. We have supplemented it in the relevant sections of our manuscript. The revised content is as follows:

Result of bootstrap showed that all coefficients in our final model were within the 95% confidence interval of the corresponding predicted value (Supplement 2). (Clean version: Line 193; Tracked version: Line 194).

We used the bootstrap method to construct a validation cohort and verify the multiple linear regression model. (Clean version: Line 353; Tracked version: Line 356).

Supplement 2. Bootstrap method to verify the multiple linear regression model

Variable

Coefficient (Bootstrap)

Std.Error

Bootstrap P value

95% Confidence Interval

Coefficient (Model)

Lower

Upper

Age

0.077

0.022

0.003

0.037

0.123

0.049

ALT

-0.003

0.006

0.554

-0.016

0.010

0.007

TBIL

0.012

0.004

0.002

0.006

0.021

0.010

ALB

-0.034

0.046

0.456

-0.125

0.058

-0.100

GGT

-0.004

0.001

0.010

-0.007

-0.002

-0.004

Constant value

1.827

2.433

0.449

-3.242

6.418

5.195

Dependent variable: voriconazole trough concentration;

Based on 1000 Bootstrap samples

Point 3: The analytical method reported for TDM based on a two-dimensional chromatography is quite uncommon and might be a source of variability of results. Do the laboratory perform proficiency testing to test their performance?

Response 3: Thank you for pointing this out. Yes, our laboratory has performed proficiency testing to test their performance. According to your suggestion, we added the results of methodological validation to illustrate the excellent performance of this two-dimensional chromatography method. In addition, our laboratory has passed the external quality assessment, which can ensure the performance of measurement. Meanwhile, the research on VRZ[1] reported by us recently also uses this analytical method. The revised content is as follows:

The intra-day and inter-day precisions were 1.94%-2.22% and 2.15%-6.78%. The absolute and relative recovery ranged from 88.2% to 93.6% and 94.2% to 105.3%, respectively. The stability of blood sample at room temperature for 8 hr and at −20°C of three repeated freeze-thaw cycles was within ±8% and ±10%, respectively. In addition, our laboratory has passed the external quality assessment, which can ensure the performance of measurement. (Clean version: Line 311; Tracked version: Line 314)

[1]        Zhao Y, Hou J, Xiao Y, et al. Predictors of Voriconazole Trough Concentrations in Patients with Child-Pugh Class C Cirrhosis: A Prospective Study [J]. Antibiotics (Basel), 2021, 10(9):

Reviewer 2 Report

Dear authors:

Line 129: it is not clear between which groups there are differences. Also in the figure, "p value < 0.001", is not clear among which groups it is referred. Too aspecific

Line 136: the reason for high SD in infectious disease patients is not very clear, please explain

Line 207: specify which particular pharmacokinetic characteristics

Line 246: specify in the same mode as line 238-239 

Line 297: ref. 45. The analytical method used: from reference, is not clear if it's a analytical method developed and validated "home made" or if it's a industrial analytical method with specifical kit application

Author Response

Thank you very much for these insightful comments. We will appreciate this opportunity to share our work and have revised the manuscript carefully. According to your suggestion, we have substantially revised the manuscript as follows.

Point 1: Line 129: it is not clear between which groups there are differences. Also in the figure, "p value < 0.001", is not clear among which groups it is referred. Too aspecific

Response 1: Thank you very much for your careful review and valuable advice. According to your suggestion, we have supplemented the methods and references in the relevant sections of our manuscript. The urological organ transplantation department was used as the reference. The revised content is as follows:

Compared with the urological organ transplantation department, the concentration distributions in the hematology (P < 0.001), infectious diseases (P < 0.001), pneumology (P < 0.001), emergency (P < 0.001), and general surgery (P = 0.035) departments were significantly different, only the pediatric department (P = 1.000) was not. (Clean version: Line 133; Tracked version: Line 134).

We used the Bonferroni method to compare distributions across each pair of the department, and the urological organ transplantation department was used as the reference. (Clean version: Line 340; Tracked version: Line 343).

Point 2: Line 136: the reason for high SD in infectious disease patients is not very clear, please explain

Response 2: Thank you for pointing this out. It has been explained accordingly. The reason may be that infectious disease patients were mostly accompanied by chronic hepatitis B, hepatitis A, alcoholic hepatitis, etc. There was a great interindividual variability in their liver function. Voriconazole is metabolized by drug metabolizing enzyme CYP2C19 [8, 9], and its concentration has a great correlation with liver function [10, 11]. We have added this part to the discussion section. The revised content is as follows:

Meanwhile, the reason for high SD in infectious disease patients may be that these patients were mostly accompanied by chronic hepatitis B, hepatitis A, alcoholic hepatitis, etc. There was a great interindividual variability in their liver function. Voriconazole is metabolized by drug metabolizing enzyme CYP2C19 [8, 9], and its concentration has a great correlation with liver function [10, 11]. (Clean version: Line 258; Tracked version: Line 261).

Point 3: Line 207: specify which particular pharmacokinetic characteristics

Response 3: Thank you so much for your careful review. It has been specified accordingly. The revised content is as follows:

The other reason may be differing pharmacokinetic characteristics such as the clearance and the volume of distribution among populations [36, 37], which were reported to have different influence factors such as platelet count, age and CYP2C19 phenotype. (Clean version: Line 211; Tracked version: Line 212).

Point 4: Line 246: specify in the same mode as line 238-239

Response 4: Thank you very much for your careful review and valuable advice. We have amended the manuscript accordingly. The revised content is as follows:

Only 14.0% (118/843) infectious diseases department samples were collected with a maintenance dose of 200 mg q12h, since VRZ is generally given at 100 or 200 mg qd. (Clean version: Line 253; Tracked version: Line 254).

Point 5: Line 297: ref. 45. The analytical method used: from reference, is not clear if it's a analytical method developed and validated "home made" or if it's a industrial analytical method with specifical kit application

Response 5: Thank you for pointing this out and we are sorry for the confusion, it has been modified. The analytical method used was developed and validated by ourselves before. The article previously published by us describing detailed chromatographic conditions and proficiency testing results [45] was cited here to illustrate the excellent performance of this analytical method. According to your suggestion, we added some detailed results of methodological validation. The revised content is as follows:

We used a fully validated automatic two-dimensional high-performance liquid chromatography technique (Demeter Instrument Co., Ltd., Hunan, China) developed and validated by ourselves before to measure VRZ concentration. (Clean version: Line 306; Tracked version: Line 311)

The intra-day and inter-day precisions were 1.94%-2.22% and 2.15%-6.78%. The absolute and relative recovery ranged from 88.2% to 93.6% and 94.2% to 105.3%, respectively. The stability of blood sample at room temperature for 8 hr and at −20°C of three repeated freeze-thaw cycles was within ±8% and ±10%, respectively. In addition, our laboratory has passed the external quality assessment, which can ensure the performance of measurement. (Clean version: Line 311; Tracked version: Line 314)

  1. Wang, T., Chen, S., Sun, J., Cai, J., Cheng, X., Dong, H., Wang, X., Xing, J., Dong, W., Yao, H., et al. Identification of factors influencing the pharmacokinetics of voriconazole and the optimization of dosage regimens based on Monte Carlo simulation in patients with invasive fungal infections [J]. J Antimicrob Chemother 2014, 69(2): 463-470.
  2. Lee, S., Kim, B.H., Nam, W.S., Yoon, S.H., Cho, J.Y., Shin, S.G., Jang, I.J., Yu, K.S. Effect of CYP2C19 polymorphism on the pharmacokinetics of voriconazole after single and multiple doses in healthy volunteers [J]. J Clin Pharmacol 2012, 52(2): 195-203.
  3. Pascual, A., Calandra, T., Bolay, S., Buclin, T., Bille, J., Marchetti, O. Voriconazole therapeutic drug monitoring in patients with invasive mycoses improves efficacy and safety outcomes [J]. Clin Infect Dis 2008, 46(2): 201-211.
  4. Boyd, A.E., Simon, M., Howard, S.J., Moore, C.B., Keevil, B.G., Denning, D.W. Adverse reactions to voriconazole [J]. Clinical Infectious Diseases 2004, 39(8): 1241-1244.
  5. Tang, D., Song, B.L., Yan, M., Zou, J.J., Zhang, M., Zhou, H.Y., Wang, F., Xiao, Y.W., Xu, P., Zhang, B.K., et al. Identifying factors affecting the pharmacokinetics of voriconazole in patients with liver dysfunction: A population pharmacokinetic approach [J]. Basic Clin Pharmacol Toxicol 2019, 125(1): 34-43.
  6. Liu, Y., Qiu, T., Liu, Y., Wang, J., Hu, K., Bao, F., Zhang, C. Model-based Voriconazole Dose Optimization in Chinese Adult Patients With Hematologic Malignancies [J]. Clin Ther 2019, 41(6): 1151-1163.

Round 2

Reviewer 1 Report

The authors have addressed all the suggestions and comments by the reviewer and can now be considered acceptable in its present form

Reviewer 2 Report

Dear Authors,

you have responded to all requested revisions

Kind regards